# Redox State and Lysosomal Activity in Women with Ovarian Cancer with Tumor Recurrence and Multiorgan Metastasis

**DOI:** 10.3390/molecules26134039

**Published:** 2021-07-01

**Authors:** Paweł Sutkowy, Jolanta Czuczejko, Bogdan Małkowski, Karolina Szewczyk-Golec, Rita Łopatto, Marta Maruszak, Alina Woźniak

**Affiliations:** 1Department of Medical Biology and Biochemistry, Ludwik Rydygier Collegium Medicum in Bydgoszcz, Nicolaus Copernicus University in Toruń, 85-092 Bydgoszcz, Poland; karosz@cm.umk.pl (K.S.-G.); al1103@cm.umk.pl (A.W.); 2Department of Psychiatry, Ludwik Rydygier Collegium Medicum in Bydgoszcz, Nicolaus Copernicus University in Toruń, 85-094 Bydgoszcz, Poland; joczu@cm.umk.pl; 3Department of Nuclear Medicine, Oncology Center in Bydgoszcz, 85-796 Bydgoszcz, Poland; rita.lopatto@gmail.com (R.Ł.); m.maruszak@gmail.com (M.M.); 4Department of Imaging Diagnostics, Ludwik Rydygier Collegium Medicum in Bydgoszcz, Nicolaus Copernicus University in Toruń, 85-794 Bydgoszcz, Poland; malkowskib@cm.umk.pl

**Keywords:** oxidative stress, lysosomal enzymes, cancer

## Abstract

The aim of the study is to evaluate oxidant–antioxidant balance as well as lysosomal and anti-protease activities in ovarian cancer since it has been emphasized that the crucial inducing factor of carcinogenesis may be reactive oxygen/nitrogen species or, more precisely, oxidative stress-induced inflammation. The study involved 15 women with ovarian cancer, aged 59.9 ± 7.8 years, and 9 healthy women aged 56.3 ± 4.3 years (controls). The study material was venous blood collected from fasting subjects. In erythrocytes, the activities of superoxide dismutase, glutathione peroxidase, and catalase, as well as concentrations of conjugated dienes (CDs) and thiobarbituric acid reactive substances (TBARS), were investigated. CD, TBARS, and vitamins A and E plasma concentrations were also determined. Moreover, total antioxidant capacity and concentrations of 4-hydroxynonenal adducts and 8-iso-prostaglandin F2α, as well as activities of acid phosphatase, arylsulfatase, cathepsin D, and α_1_-antitrypsin, were studied in serum. The vitamin E and 8-iso-prostaglandin F2α concentrations as well as arylsulfatase activity were lower in the women with cancer compared to the controls (*p* = 0.006, *p* = 0.03, *p* = 0.001, respectively). In contrast, cathepsin D activity was lower in the controls (*p* = 0.04). In the peripheral blood of the women with cancer, oxidant–antioxidant and lysosomal disturbances were observed.

## 1. Introduction

Transformation of normal cells into neoplastic cells is connected with cell cycle disturbances induced by mutations of proto-oncogenes (activation of cell divisions) and/or suppressor genes (blocking of cell divisions) as well as mutator genes (DNA protection against damage or repairing damaged DNA). It is a multistage and prolonged process defined by at least three stages: initiation, promotion, and progression. Increasingly, more data have emphasized that one of the most important direct cancer initiation factors are reactive oxygen/nitrogen species (ROS/RNS), actually oxidative-stress-induced inflammation. Persistent cancer stimulus is usually a direct or indirect source of oxidative stress leading to DNA damage, genome instability, and the modulation of signaling pathways (especially NF-κB and MAPK) and, thus, the promotion of neoplastic transformation. Such stimulus is too weak to cause apoptosis; however, it is sufficient to disturb cell functions. As a consequence, chronic inflammation appears at the site of damage, which is mediated by inflammatory cells (mast cells and leukocytes). The cells are the next significant source of ROS/RNS (respiratory burst). They also release many free mediators of inflammation (e.g., metabolites of arachidonic acid, cytokines, chemokines) that augment the recruitment of other immunological cells to the site of damage and intensify free radical generation. Thus, the chronic harmful stimulus and long-term irritation of the same place lead to a vicious circle that can result in irreversible dysfunction of tissue/organ or chronic disease, including cancer. Elevated ROS concentrations are found in cancer cells not only in the initiation stage but also during promotion and progression when tumor metastasis can occur [1]. ROS/RNS are oxidant molecules that are necessary for the normal function of the organism; in relatively high levels, they are necessary for the maintenance of the high metabolic activity of tumors as well as for their growing and formation in distant places (metastasis) [2]. However, when their concentrations exceed antioxidant abilities, both in normal and neoplastic cells, they become a source of oxidative stress, resulting in serious damage of cellular structures (proteins, sugars, lipids, DNA, and others), which leads to apoptosis or even necrosis. Cancer can adapt to excessive ROS levels by increasing its antioxidant capacity. It was revealed that upregulated antioxidant defense is necessary for metastasis since too high ROS concentrations induce cancer regression or limit its development and completely block metastasis [1,2]. In contrast, other clinical data have revealed that antioxidant supplementation as a therapeutic anti-cancer strategy not only failed to benefit women but was associated with a significant increase in cancer incidence [2]. Thus, cancer needs for its proper development a specific redox state, characterized by relatively high ROS/RNS levels accompanied by increased antioxidant capacity. The issue is still not fully understood and is very important in the search area of new anti-cancer strategies.

Determination of the antioxidant potential and levels of oxidative stress products provides an idea about ROS levels and the state of redox equilibrium (oxidants vs. antioxidants). This can be established by determining the activities of antioxidant enzymes (high molecular weight antioxidants, e.g., superoxide dismutase, SOD, glutathione peroxidase, GPx, catalase, and CAT), the concentration of antioxidant vitamins (low molecular weight antioxidants, e.g., vitamins A and E), and total antioxidant capacity (TAC) in tissue homogenate or body fluid [3]. Moreover, oxidative stress can affect lysosomal activity, i.e., releasing enzymes by lysosomes into intra- and extracellular spaces [4]. Therefore, the aim of the study is to determine the levels of the aforementioned antioxidants and TAC as well as the levels of oxidative stress products and the activities of selected lysosomal enzymes, together with serine protease inhibitors (α_1_-antitripsin, AAT) activity in venous blood, in women with ovarian cancer.

## 2. Material and Methods

### 2.1. Study Subjects

The study included two groups: women with ovarian cancer of stage IV on the FIGO scale (59.9 ± 7.8 years; *N* = 15) who were patients of an oncology center in Bydgoszcz, Poland, and a control group consisting of completely healthy women of similar age (56.3 ± 4.3, *N* = 9) who had never suffered from any neoplastic disease. The most common morphological type of cancer (epithelial) was diagnosed in each patient in positron emission tomography (PET) combined with computed tomography (CT) using the ^18^F fluorine isotope labeled with deoxyglucose (FDG) as a marker (FDG PET/CT examination). The anti-cancer therapy consisted of surgical procedure (ovariectomy) and/or chemotherapy combined with glucocorticosteroid treatment. However, 6–12 months after the therapy, recurrences and multiorgan tumor metastases in the abdominal cavity were diagnosed in each patient (FDG PET/CT examination). The cancer had spread to the pelvis, the peritoneum, the liver and its axillary, and thoracic lymph nodes. Thus, each patient that participated in the study had epithelial ovarian cancer of the same highest stage. Moreover, the patients had no coexisting diseases (e.g., diabetes, atherosclerosis), and their dietary habits were “typical” (no vegans or vegetarians, similar to the control group). The total number of such patients in the hospital (the centre of oncology) during the study period (one year) was 18; however, because of too big a difference in age compared to the control group, three patients had to be excluded. More features of the examined women are presented in Table 1.

The material for the study was fasting peripheral blood collected from the subjects—in women with ovarian cancer, half an hour before their FDG PET/CT examination. The blood was taken from the basilic vein (elbow flexion) into two vacuum tubes: potassium versenate (K_2_EDTA) for plasma (4 mL) and with clot activator and separation gel for serum (5 mL).

This experimental protocol did not carry any additional risk to the health of the women with ovarian cancer because the blood was taken once during blood collection for PET/CT examination, which is a standard medical procedure for cancer patients in that hospital.

Directly prior to study commencement, the women of both groups voluntarily signed a consent form for participation in the study. All participants were informed of the purpose of the study. The study was approved by the Bioethics Committee at Collegium Medicum of Nicolaus Copernicus University in Bydgoszcz, Poland (no. KB 363/2017), and has been registered in the clinical trials database (ClinicalTrals.gov, no. NCT03470857). The authors confirm that all methods in the study were performed in accordance with relevant guidelines and regulations.

### 2.2. Methods

The activities of SOD, GPx, and CAT were assessed in erythrocytes; in contrast, the concentrations of vitamins A and E were assessed in plasma. The concentrations of oxidative stress products were determined in erythrocytes, plasma, and serum: conjugated dienes and thiobarbituric acid reactive substances in plasma/erythrocytes (CDpl/er and TBARSpl/er, respectively); the concentrations of 4-hydroxynonenal (HNE) adducts, 8-iso-prostaglandin F2α (8-iso-PGF2α), and protein carbonyls in serum. In blood serum, TAC and the activities of lysosomal enzymes (acid phosphatase, AcP, arylsulphatase, ASA and cathepsin D, CTS D) as well as AAT activity were also measured. The methods of the study were based on spectrophotometric measurements of absorbance.

The erythrocytic activities of antioxidant enzymes were determined in a suspension of red blood cells with a hematocrit index of 50% (1:1, *v/v*, erythrocyte in phosphate-buffered saline, PBS) (the same suspension was used for CDer and TBARSer assays). SOD (copper-zinc superoxide dismutase, SOD-1, E.C. 1.15.1.1) activity was assayed according to Misra and Fridovich’s method [5]. The principle of the method is the inhibition reaction of adrenaline oxidation to adrenochrome catalyzed by the enzyme in an alkaline environment. SOD activity is shown as the enzyme unit per gram of hemoglobin (U/g Hb). Peroxidase activity in erythrocytes is expressed as the ability of the GPx (E.C. 1.11.1.9) to reduce hydrogen peroxide using glutathione (GSH). Oxidized glutathione (GSSG) is reduced by glutathione reductase (GR) in the assay. The cofactor of the reaction is reduced nicotinamide adenine dinucleotide phosphate (NADPH), which is transformed into an oxidized form, resulting in the change in absorbance of the sample solution [6]. The enzyme activity is expressed similarly to SOD activity. The determination of CAT activity was based on a decrease in absorbance in the solution of the erythrocyte suspension and hydrogen peroxide [7]. CAT activity is shown as the international enzyme unit per g Hb (IU/g Hb).

The plasma vitamins A and E concentrations were assayed using high-performance liquid chromatography (HPLC). The procedure consisted of the following steps: denaturation of plasma proteins with 800 μL acetonitrile, centrifugation, collection of supernatant and its filtering, and the extraction of the product with hexane. Next, the hexane fraction, which contained the vitamins, was evaporated to dryness under a nitrogen atmosphere at 40 °C and mixed using ultrasounds with 100 μL of acetonitrile + ethanol solution. Such prepared samples were analyzed in an HPLC column (C18, l = 250 mm) at the wavelength of 292 nm. The vitamin concentrations are expressed in micrograms per liter of plasma (μg/L).

Serum TAC was measured using a commercial assay kit. The TAC assay is based on the reduction of copper (II) to copper (I) ions by the antioxidant uric acid (kit reagent) or the antioxidants present in the sample. The copper (I) ions react with a chromogenic reagent, resulting in a color product with a maximum absorbance at 490 nm. The net absorbance (λ equal to 490 nm) values of the studied samples were compared with the uric acid standard curve concurrently determined with the samples in a 96-well plate. The results of the TAC assay are expressed as millimoles of uric acid equivalents per liter of serum (mM UAE).

The plasma and erythrocytic concentrations of CD were assessed by adding chloroform into blood plasma or the erythrocyte suspension and collecting the lower fraction of the solution into clean tubes after centrifugation. Next, the samples were evaporated under a nitrogen atmosphere and dissolved in cyclohexane, and the absorbance of the resulting solution was read at a wavelength of 233 nm [8]. The CDpl/er concentrations are expressed as absorbance value per milliliter of plasma (Abs./mL) or absorbance value per gram of hemoglobin (Abs./g Hb), respectively.

The plasma and erythrocytic concentrations of TBARS were measured using the following reagents, which were added to blood plasma or the erythrocyte suspension: 0.375% thiobarbituric acid (TBA), 15% trichloroacetic acid (TCA), and 0.25 M HCl. The next step was the incubation of the working solutions in hot water (100 °C; water bath) for 20 min. The sample solutions were subsequently cooled down (to 4 °C) and centrifuged for 15 min. After centrifugation, supernatants were collected, and absorbance at λ equal to 532 nm was measured [9,10]. The TBARSpl/er concentrations are expressed as nanomoles of malondialdehyde per milliliter of plasma (nmol MDA/mL) or nanomoles of malondialdehyde per gram of hemoglobin (nmol MDA/g Hb), respectively.

Serum concentrations of HNE adducts, 8-iso-PGF2α, and protein carbonyls were evaluated using commercial enzyme-linked immunosorbent assays (ELISA); in the case of HNE adducts and 8-iso-PGF2α, it was competitive ELISA.

Serum samples with 4-HNE protein adducts (HNE adducts) or HNE–bovine serum albumin (BSA) standards were added to a 96-well protein binding plate preabsorbed by the HNE conjugate. After a brief incubation, an anti-HNE conjugate antibody was added, followed by an antibody conjugated with horseradish peroxidase (HRP) and the substrate for HRP. The HNE adduct concentration is inversely proportional to read absorbance, and it is determined by comparison with the HNE–BSA standard curve, which was determined in parallel with the studied samples. The HNE adduct concentration is shown as micrograms per milliliter of serum (μg/mL).

The principle of the 8-iso-PGF2α test is the use of an antibody to 8-iso-PGF2α, which was incubated in microtiter plate wells pre-coated by goat anti-rabbit antibodies. Next, 8-iso-PGF2α standards and the studied samples, both mixed with an 8-iso-PGF2α–HRP conjugate, were added to the wells. The colored reaction was started by adding the substrate for HRP. Both substrates (standard and sample 8-iso-PGF2α) compete for binding to the antibody bound to the plate; hence, the concentration of 8-iso-PGF2α in the samples is inversely proportional to the amount of 8-iso-PGF2α–HRP conjugate. The concentration was read using a standard curve and is expressed in the paper in picograms per milliliter of serum (pg/mL).

The concentration of protein carbonyls was measured by comparing absorbance values for serum samples with those of known standard curves created by reduced/oxidized BSA standards. In this ELISA kit, specific antibodies and HRP conjugated with secondary antibodies were used as well. The concentration of protein carbonyls was directly proportional to the absorbance. The concentration is expressed in nanomoles of oxidized proteins per milligram of total protein in the serum sample (nmol/mg).

The serum activity of AcP (E.C. 3.1.3.2) was determined using Bessey’s method, according to Krawczyński’s modification [11]. For this purpose, disodium p-nitrophenylphosphate (substrate) in 0.5 mol/L citrate-tartrate-formaldehyde buffer at pH 4.9 was used. The measure of enzyme activity was the amount of 4-nitrophenol (4-NP) released during the enzymatic hydrolysis of the substrate. Hence, the activity is shown as nmol 4-NP/mg protein/min.

Determination of serum ASA activity (E.C. 3.1.6.1) was performed using Roy’s method, as modified by Błeszyński [12]. To determine ASA activity, 0.01 mol/L 4-nitrocatechol sulfate (substrate) in 0.5 mol/L acetate buffer at pH 5.6 was used. The measure of enzyme activity was the amount of 4-nitrocatechol (4-NC) released during the enzymatic hydrolysis of the substrate; therefore, the activity is shown as nmol 4-NC/mg protein/min.

To determine serum CTS D activity (E.C. 3.4.23.5), 2% denatured bovine hemoglobin was subjected to hydrolysis, catalyzed by the enzyme at 37 °C. The reaction was subsequently stopped with a 0.1 mol/L NaOH solution, and a phenolic reagent was added (color reaction). The absorbance was measured at λ equal to 660 nm. CTS D activity is expressed as nmol tyrosine per mg protein per minute (TYR/mg/min) since the assay results were extrapolated from the standard curve of tyrosine activity [13].

Determination of serum AAT activity was performed based on the measurement of the decreasing enzymatic activity of trypsin (TR) after a short incubation with the studied serum [14]. The enzyme activity is shown as mg TR/mL serum.

### 2.3. Statistical Analysis

The results were statistically analyzed using a parametric test—Student’s *t*-test for independent samples. Assumptions of the test were verified using the Shapiro–Wilk test and Levene’s test. Moreover, Pearson’s linear correlation coefficients were calculated between investigated parameters. The results are expressed as an arithmetic mean ± standard deviation. Differences between the means were considered statistically significant at *p* < 0.05.

## 3. Results

All study results are shown in Table 2. Among antioxidants and TAC, the vitamin E concentration was statistically significantly higher (by 42.7%) in the healthy study participants (control) than in the women with cancer (*p* < 0.01). The vitamin A concentration was also higher in the healthy controls but at a statistically insignificant level (24.4% difference, *p* > 0.05). Other antioxidant levels changed in a statistically insignificant manner as well. The activity of GPx was insignificantly higher in healthy controls, while the SOD and CAT activities, as well as TAC, were slightly higher in the women with ovarian cancer (*p* > 0.05). The concentrations of oxidative stress markers were similar in both groups (*p* > 0.05), excluding the 8-iso-PGF2α concentration. It was higher by 23.8 pg/mL serum in healthy controls (38.3% difference, *p* < 0.05).

The activity of ASA was higher by 41% in the healthy women (*p* < 0.01), whereas CTS D activity was higher in the women with cancer (difference of 27.7%, *p* < 0.05). The activity of AcP was slightly higher in the controls (by 22.4%; *p* > 0.05) in contrast with AAT activity that was higher in the women with cancer (23.9% difference; *p* > 0.05).

The following linear correlations were revealed in the study on women with ovarian cancer: GPx activity was correlated with TBARSpl concentration (*r* = −0.96, *p* = 0.042); vit. A concentration with TAC (*r* = −0.98, *p* = 0.018), CDpl concentration (*r* = −0.95, *p* = 0.048), 8-iso-PGF2α concentration (*r* = −0.99, *p* = 0.012; Figure 1), and ASA activity (*r* = −0.95, *p* = 0.049); while TAC was linearly correlated with CDpl and 8-iso-PGF2α concentrations (*r* = 0.97, *p* = 0.032 and *r* = 0.96, *p* = 0.035, respectively). The concentrations of CDpl and 8-iso-PGF2α (*r* = 0.97, *p* = 0.026; Figure 2) as well as the concentration of 8-iso-PGF2α vs. ASA activity (*r* = 0.96, *p* = 0.035) were also linearly correlated.

In the group of healthy control subjects, statistically significant Pearson’s correlations were revealed between GPx activity and CDpl concentration (*r* = −0.96, *p* = 0.044) as well as between vitamin E concentration and AAT activity (*r* = 0.98, *p* = 0.020).

## 4. Discussion

The study showed that the levels of antioxidant vitamins in serum in women with ovarian cancer were lower than in healthy women of the control group (especially vitamin E, *p* = 0.006). It indicates higher vitamin consumption in women with cancer as a result of an increased generation of ROS/RNS; since these low molecular weight antioxidants are not synthesized in the organism, they have to be supplied with food, and both study groups were required to fast before blood collection. The vitamins had to “wear out” due to the increased production of free radicals [15]. Moreover, intensified ROS/RNS generation in the women with cancer may be confirmed with very strong linear correlations that were noted in serum and plasma—first, two negative corrections between vitamin A concentration and the concentrations of CDpl (*r* = −0.95 at *p*-level of 0.048) and 8-iso-PGF2α (*r* = −0.99 at *p*-level of 0.012; Figure 1); second, one positive correlation between the concentrations of CDpl and 8-iso-PGF2α (*r* = 0.97 with *p* = 0.026; Figure 2)—since the latter are primary (CDpl) and secondary (8-iso-PGF2α, type of F2-isoprostane) products of free-radical-mediated lipid peroxidation [16]. High concentrations of free radicals are a cancer-initiating factor (oxidative damage of DNA, proteins, and lipids); they are needed to maintain a high rate of tumor metabolism, its growing, angiogenesis, and metastasis [1]. H_2_O_2_ seems to be a particularly important free radical because of the increased activity of NF-κB, one of the most crucial factors in carcinogenesis mediated by ROS-induced inflammation [1], which is connected with the overexpression of SOD and the decreased activity of CAT [17]. SOD is responsible for generating hydrogen peroxide as a result of removing superoxide anion radicals, while CAT decomposes this toxic compound (H_2_O_2_) [3]. It was demonstrated in the study that erythrocytic activities of both enzymes in the women with ovarian cancer were statistically insignificantly higher than in the control group (especially SOD), which may suggest a slightly higher production of free radicals in these women. Furthermore, a very strong and negative correlation between vitamin A concentration and TAC (*r* = −0.98 at *p* = 0.018), found in the study in the women with ovarian cancer, may indicate that this hydrophobic antioxidant did not have a significant influence on the TAC of patients’ peripheral blood, perhaps in favor of endogenous hydrophilic antioxidants (possibly uric acid, which is formed after the disintegration of purines, the major antioxidant in human blood plasma [18]). Possibly, it resulted from the depletion of hydrophobic vitamins as a result of excessive ROS generation in the women, as already described. The concentrations of CDpl and TBARSpl and the serum concentration of protein carbonyls, successive markers of oxidative stress [16], were also slightly higher in the women with cancer compared to the healthy ones (*p* > 0.05). In contrast, the concentration of protein HNE adducts was lower in the women with cancer, whereas TAC was higher (*p* > 0.05). However, these differences were at a statistically insignificant level, while the serum concentration of 8-iso-PGF2α in the women with ovarian cancer was statistically significantly lower compared to the healthy control women (*p* = 0.03). 8-iso-PGF2α, similar to HNE, is the marker of oxidative-stress-induced lipid peroxidation [3,16]. This result may point to upregulated antioxidant mechanisms in peripheral blood in the women suffering from ovarian cancer. Therefore, in general, it can be concluded that elevated levels of ROS/RNS in women with cancer are associated with concurrently enhanced antioxidant mechanisms, primarily in the serum of their peripheral blood. Furthermore, oxidative stress cannot be stated since this is a condition characterized by a clear predominance of oxidation products over products of antioxidation, which is manifested by damage to cell structures [1]. The literature confirms such reports. The higher the concentration of free radicals in the tumor, the higher the antioxidant capacity needs to be so that the tumor can grow properly and generate metastases [2]. For instance, elevated glutathione (GSH) levels (the most important soluble antioxidant inside the cell) are observed in various types of cancer cells and solid tumors. This even leads to the increased resistance of neoplastic cells/tissues to chemotherapy and induces radioresistance since increased antioxidant capacity induces the hyposensitivity of cancer cells to ROS [19]. The correlations obtained between the TAC vs. the CDpl and 8-iso-PGF2α concentrations can indicate this specific oxidoreduction state, which was found in the women with ovarian cancer in the study (*r* = 0.97 with *p* = 0.032 and *r* = 0.96 with *p* = 0.035, respectively) since the correlations inform us that an increased level of ROS/RNS is accompanied by increased antioxidant capacity. In the study, an almost fully negative linear correlation between GPx activity and TBARSpl concentration was also revealed in the women with cancer (*r* = −0.96, *p* = 0.042). GPx activity was also very similarly correlated with CDpl concentration in the group of healthy women (*r* = −0.96, *p* = 0.044). These relationships are difficult to explain. It can only be stated that in this case, the oxidation–reduction relationship in peripheral blood is the same as in the women with ovarian cancer and the healthy women since this is a correlation between antioxidant GPx activity in erythrocytes and ROS-mediated lipid peroxidation products in plasma.

Moreover, statistically and significantly lower serum ASA activity in the women with ovarian cancer was noted in the study compared to the healthy controls (*p* = 0.001). The lower ASA activity may refer to intensified ROS/RNS formation because ASA is activated by the formylglycine-generating enzyme and the enzyme is inhibited in the process, which is mediated by free radicals [20]. In the study, very strongly and positively correlated ASA activity with 8-iso-PGF2α concentration (*r* = 0.96, *p* = 0.035) may confirm this thesis if the vitamin E results are taken into account since the lack of free-radical-mediated increase of permeability of lysosomal membranes was possibly observed. Oxidative modulations of lipids in lysosomal membranes may potentiate the release of lysosomal enzymes [21], while, as it was mentioned, 8-iso-PGF2α is the product of such changes [16]. Vitamin E could prevent it in the study patients. The negative correlation between vitamin A concentration and ASA activity, which was demonstrated in the women with ovarian cancer, may be associated with this as well (*r* = −0.95, *p* = 0.049); however, it suggests higher ROS concentrations (the worn-out vitamin does not stop lipoperoxidation).

In the case of AcP activity in both study groups, similar values were shown. This is inconsistent with the literature data since serum activity of the enzyme is found to be higher in the course of neoplastic disease in humans [22]. The enzyme may also be the source of ROS in the Fenton reaction due to the iron ions included in its structure [23]. In contrast to ASA, AcP activity in the women with ovarian cancer was insignificantly lower compared to the healthy controls (*p* > 0.05). However, it is worth mentioning that lysosomal hydrolases may be selectively released from cellular organelles [24]. In turn, CTS D activity was relatively high in the women with ovarian cancer (*p* = 0.04). The literature confirms such a phenomenon. Cathepsin D is the major proteolytic hydrolase of lysosomes. It is involved in, i.e., the proteolytic transformation of foreign antigens and prohormones and is released outside the cell in inflammation sites and during tumor progression [25]. In general, the lysosomes are the organelles that release their hydrolases as a consequence of the action of numerous exo- and endogenous cell death factors, including excessive ROS/RNS concentrations [26], based on the aforementioned mechanism (increased permeabilization of lysosomal membranes induced by oxygen-radical-mediated peroxidation of their lipids [21]). In contrast, for AAT, the serum activity of this protein was higher in the women with cancer compared to the controls. AAT is an anti-inflammatory protein, and it has the highest anti-protease activity in venous blood [27]. Most likely, it also has antioxidant properties [28]. The literature reports inform us that enzyme activity can be inhibited by ROS [29]; thus, it may suggest the opposite conclusion regarding intensified ROS production in the women with ovarian cancer. First, however, the difference was statistically insignificant. Second, the literature also provides information that liposoluble vitamins (E and A) may prevent this inhibition [29], and, as it was mentioned, the serum vitamin E concentration was relatively low in the women with cancer (*p* = 0.006). Therefore, the mentioned free-radical-mediated consumption of vitamin E could contribute to the slightly higher activity of AAT in the women with cancer. The described mechanism may be confirmed by the positive and very strong linear correlation between vitamin E concentration and AAT activity found in the healthy women group, and this may indicate a significant relationship between both parameters in redox equilibrium (*r* = 0.98, *p* = 0.020). In general, the study results of the described lysosomal enzymes correspond with results obtained for the oxidant–antioxidant equilibrium.

## 5. Conclusions

In the venous blood of women with ovarian cancer with tumor recurrence and multiorgan metastasis, elevated oxidative potential, probably together with increased antioxidant capacity, was observed. The results indicate oxidant–antioxidant and lysosomal disturbances.

## Figures and Tables

**Figure 1 molecules-26-04039-f001:**
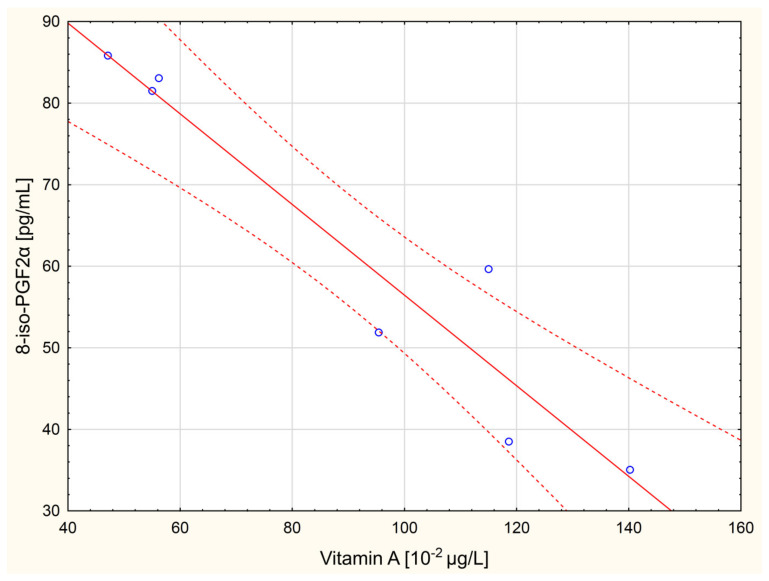
Linear correlation between the concentrations of vitamin A and 8-iso-prostaglandin F2α in serum of venous blood in women with ovarian cancer (*r* = −0.99, *p* = 0.012).

**Figure 2 molecules-26-04039-f002:**
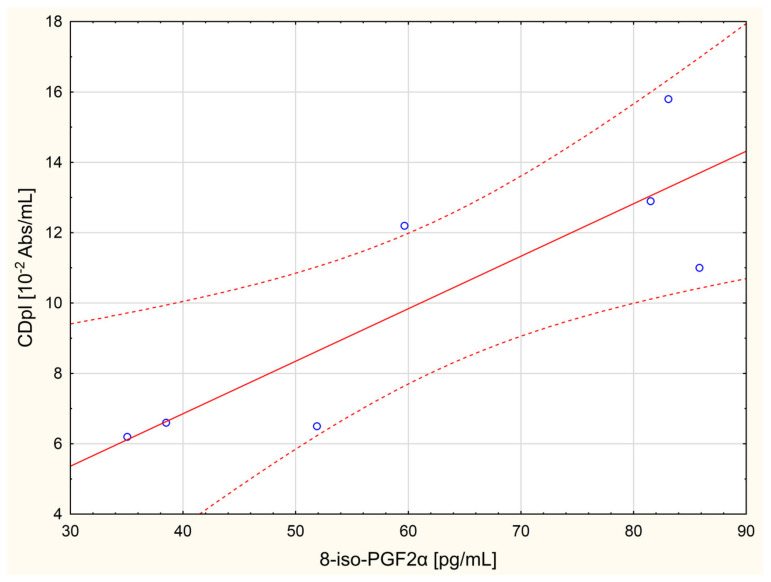
Pearson’s correlation between the serum concentration of 8-iso-prostaglandin F2α and the plasma concentration of conjugated dienes in women with ovarian cancer (*r* = 0.97, *p* = 0.026).

**Table 1 molecules-26-04039-t001:** Characteristics of the investigated women (arithmetic mean ± standard deviation).

	Women with Ovarian Cancer	Healthy Controls
Number of subjects	15	9
Age [years]	59.9 ± 7.8	56.3 ± 4.3
Body weight [kg]	76.9 ± 15.6	62.3 ± 10.0
Body height [cm]	163.3 ± 7.3	161.8 ± 3.4
Body mass index [kg/m^2^]	28.9 ± 6.1	23.8 ± 3.7

**Table 2 molecules-26-04039-t002:** Oxidant–antioxidant balance and activities of selected lysosomal enzymes, together with anti-protease activity in peripheral blood, in women with ovarian cancer.

	Women with Cancer (*N* = 15)	Healthy Controls (*N* = 9)
Antioxidants and TAC		
SOD [10 U/g Hb]	83.0 ± 17.9	78.9 ± 8.6
GPx [10^–1^ U/g Hb]	61.2 ± 21.4	81.6 ± 21.7
CAT [IU/g Hb]	66.0 ± 11.3	65.4 ± 10.4
Vitamin A [10^−2^ μg/L]	98.6 ± 32.9	122.7 ± 14.7
Vitamin E [μg/L]	66.9 ± 20.3	95.5 ± 20.2 *
TAC [10^−2^ mM UAE]	67.0 ± 16.0	57.4 ± 12.9
Oxidative stress products		
CDpl [10^−2^ Abs/mL]	10.6 ± 3.3	9.4 ± 2.8
CDer [10^−2^ Abs/g Hb]	10.1 ± 3.1	12.6 ± 3.9
TBARSpl [10^−2^ nmol MDA/mL]	55.2 ± 12.5	49.7 ± 3.4
TBARSer [nmol MDA/g Hb]	27.5 ± 8.7	21.5 ± 6.9
HNE adducts [10^−1^ μg/mL]	6.2 ± 1.9	7.6 ± 1.9
8-iso-PGF2α [pg/mL]	62.2 ± 21.5	86.0 ± 1.8 *
Protein carbonyls [10^−1^ nmol/mg]	53.5 ± 14.5	48.6 ± 13.8
Lysosomal enzymes and AAT		
AcP [10^−^^4^ nmol 4-NP/mg/min]	9.8 ± 3.1	12.0 ± 2.3
ASA [10^−4^ nmol 4-NC/mg/min]	6.6 ± 1.6	9.3 ± 1.9 *
CTS D [10^−2^ nmol TYR/mg/min]	10.6 ± 2.7	8.3 ± 2.2 *
AAT [10^−1^ mg TR/mL]	11.4 ± 3.6	9.2 ± 1.1

Results are shown as arithmetic mean ± standard deviation. * Statistically significant differences compared to the women with cancer: *p* = 0.006, *p* = 0.03, *p* = 0.001, *p* = 0.04, respectively. SOD: superoxide dismutase; GPx: glutathione peroxidase; CAT: catalase; Hb: hemoglobin; TAC: total antioxidant capacity; UAE: uric acid equivalent; CDpl/er: conjugated dienes in plasma/erythrocytes; Abs.: absorbance; TBARSpl/er: thiobarbituric acid reactive substances in plasma/erythrocytes; MDA: malondialdehyde; HNE: 4-hydroxynonenal; 8-iso-PGF2α: 8-iso-prostaglandin F2α; AcP: acid phosphatase; 4-NP: 4-nitrophenol; ASA: arylsulphatase; 4-NC: 4-nitrocatechol; CTS D: cathepsin D; TYR: tyrosine; AAT: α_1_-antitrypsin; TR: trypsin.

## Data Availability

The study data used to support the findings of this study are included within the article.

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
