# Peer review of "Redox State and Lysosomal Activity in Women with Ovarian Cancer with Tumor Recurrence and Multiorgan Metastasis"

_molecules, 2021, doi:10.3390/molecules26134039_

Round 1

Reviewer 1 Report

The authors carried out a study involving 15 women with ovarian cancer to assess the influence of oxidative stress and lysosomal disturbances on this type of malignancy. The scientific content is not new, the study lacks innovation, but benefits from the fact that it uses samples of human origin, which gives the study a significant clinical relevance.

The study is well designed, the Introduction provides clear information and the methods are well described.  Regarding the results, the experimental parameters addressing oxidant-antioxidant balance and the activities of lysosomal enzymes in blood samples provided by ovarian cancer women and healthy women were moderately different, but in most cases not statistically different. However, the results were exhaustively discussed and the conclusions, although not showing high scientific soundness, are supported by the results.

I would, however, recommend an update of the references. In fact, among the 30 references, the most recent date from 2015 (1), 2013 (1) and 2012 (1); 12 references are from 2000 to 2010, 7 from 1990 to 1999 and 8 from 1952 to 1989.

Reviewer 2 Report

Dear Authors,

The topic of the article is interesting to me, and I believe this study would be publishable in Molecules, but there are too many serious concerns and doubts that I must recommend for publication. There is no information about the type of cancer , stage and FIGO in individual patients. No data about coexisting diseases (e.g. diabetes, atherosclerosis), no information about dietary habits of the patients (vegetarian or vegan ) and group of patients is very small.

I hope that you benefit from my observations.

Best regards,
